# A Structure-Guided Designed Small Molecule Is an Anticancer Agent and Inhibits the Apoptosis-Related MCL-1 Protein

**Ingrid V. Machado** [1], **Luiz F. N. Naves** [1], **Jean M. F. Custodio** [2,*], **Hérika D. A. Vidal** [1], **Jaqueline E. Queiroz** [1], **Allen G. Oliver** [2], **Joyce V. B. Borba** [3], **Bruno J. Neves** [3], **Lucas M. Brito** [4], **Claudia Pessoa** [4], **Hamilton B. Napolitano** [1] and **Gilberto L. B. de Aquino** [1]

1 Ciências Exatas e Tecnológicas, Universidade Estadual de Goiás, Anapolis 75132-903, GO, Brazil
2 Department of Chemistry and Biochemistry, University of Notre Dame, Notre Dame, IN 46617, USA
3 LabMol—Laboratory for Molecular Modeling and Drug Design, Faculty of Pharmacy, Universidade Federal de Goiás, Goiânia 74605-170, GO, Brazil
4 Laboratory of Experimental Oncology, Center for Research and Drug Development, Federal University of Ceará, Fortaleza 60430-275, CE, Brazil
* Correspondence: jeanmfcustodio@gmail.com

**Abstract:** Cancer resistance to chemotherapy and radiation therapies presents significant challenges, necessitating the exploration of alternative approaches. Targeting specific proteins at the molecular level, particularly their active sites, holds promise in addressing this issue. We investigated the potential of 4′-methoxy-2-nitrochalcone (MNC) as an MCL-1 inhibitor, examining its chemical and structural characteristics to elucidate its biological activity and guide the selection of potential candidates. We conducted a docking study, followed by synthesis, structural characterization, theoretical calculations, and in vitro experiments to comprehensively evaluate MNC. The docking results revealed MNC's excellent binding within the active site of MCL-1. At 50 µM, MNC demonstrated 99% inhibition of HCT116 cell proliferation, with an IC50 value of 15.18 µM after 24 h. Treatment with MNC at 30.36 and 15.18 µM resulted in reduced cell density. Notably, MNC exhibited marked cytotoxicity at concentrations of 15.58 µM and 7.79 µM, inducing high frequencies of plasma membrane rupture and apoptosis, respectively. Our findings highlight the significant biological potential of MNC as an MCL-1 inhibitor. Furthermore, we propose exploring chalcones with hydrogen bond acceptor substituents as promising candidates for studying inhibitors targeting this protein. In conclusion, our study addresses the challenge of cancer resistance by investigating MNC as an MCL-1 inhibitor. Through detailed characterization and experimental validation, we establish the efficacof MNC in inhibiting cell proliferation and inducing cytotoxic effects. These results underscore the potential of MNC as a valuable therapeutic agent and suggest the use of chalcones with hydrogen bond acceptor substituents as a basis for developing novel MCL-1 inhibitors.

**Keywords:** chalcone; MCL-1; molecular docking; Hirshfeld surface

## 1. Introduction

Cancer is among the top ten causes of death, causing 10 million deaths in 2020 [1]. By 2040, it is projected to experience an increase of over 63% [2]. According to the World Health Organization (WHO), cancer-related deaths annually outnumber those caused by HIV, malaria, and tuberculosis combined, by two and a half times [3]. The rapid division of cancer cells contributes to their aggressive growth and challenging control, leading to the formation of malignant tumors. These traits also heighten the risk of metastasis, the spread of cancer cells to other parts of the body [4]. To combat cancer, chemotherapy and radiation therapies have been employed to eliminate cancer cells [5,6]. Unfortunately, resistance to these treatments often develops among these cells [7,8].

Thus, new alternatives for cancer treatment are being studied. Some proteins have demonstrated a significant contribution to the rapid proliferation and resistance in the

treatment of these cancer cells, in which myeloid cell leukemia (MCL-1) has gained a critical prominence [9]. MCL-1 is an antiapoptotic protein that is frequently overexpressed or dysregulated in various types of cancers. Its overexpression provides cancer cells with a survival advantage by inhibiting apoptosis and promoting their continued growth and resistance to cell death signals. This makes MCL-1 an attractive target for therapeutic intervention in cancer treatment [10]. MCL-1 is a member of the Bcl-2 family of proteins which includes antiapoptotic and proapoptotic members and is the primary regulator of apoptotic processes. Hence, the balance between the relative levels of such proteins is critical for regulating the cellular process [11]. Consequently, any mechanism that interferes with this balance and failures in the normal apoptosis pathways can contribute to several diseases, such as cancer [12].

Apoptosis is controlled by protein–protein interactions between the Bcl-2 family members, subdivided into two subgroups (anti and pro) based on their structural functions and properties [13]. The antiapoptotic subgroup includes Bcl-2, BCL-xL, MCL-1, Bcl-w, and A1 and the proapoptotic subgroup includes Bax, Bak, Bad, Bid, Bim, Bik, NOXA, and PUMA [14]. Antiapoptotic proteins maintain cell survival by binding and sequestering their proapoptotic counterparts [12,15]. However, MCL-1 protects normal cells from undergoing programmed cell death and protects malignant cells when interacting with proapoptotic family Bcl-2 members [16]. Thus, MCL-1 has been recognized as a promising target in searching for new cancer therapy inhibitors [17–19].

Because chalcones have been analyzed and found to inhibit the MCL-1 [20], we investigated 4′-methoxy-2-nitrochalcone (MNC) as a new candidate. Docking simulation was performed as a starting point and indicated a good fit in the MCL-1 active site compared with the reference ligand. We present the synthesis, characterization, and crystal structure analysis of MNC. This compound was tested against the HCT116 cancer cells, which express MCL-1. Finally, theoretical calculations such as frontier molecular orbitals (FMO), molecular electrostatic potential (MEP) surface, geometry optimization, and conformational stability were carried out. Such calculations enable a better understanding of MNC structure and its binding affinity to MCL-1 and might be used to search for other candidates.

## 2. Experimental and Computational Procedures

### 2.1. Synthesis and Crystallization

The introduction of a nitro group (-NO$_2$) into the chalcone scaffold can lead to enhanced biological activity, such as improved potency and selectivity against specific target proteins. The presence of the nitro group can contribute to increased binding affinity and favorable interactions with the target, resulting in improved pharmacological effects [21,22]. Additionally, methoxy chalcones possess phenolic rings with the methoxy group, which can exhibit antioxidant effects by scavenging reactive oxygen species (ROS) and preventing oxidative damage to cells and tissues. These compounds have shown potential in modulating various cellular processes and molecular targets, making them attractive candidates for drug development and therapeutic interventions [23].

Considering the applicability of both nitro and methoxy chalcones, we propose 4′-methoxy-2-nitrochalcone (MNC) as a potential biological agent. It was synthesized through the Claisen–Schmidt aldolic condensation reaction, in which the equimolar reaction occurred between 0.450 g (3.0 mmol) of 4′-methoxyacetophenone (Sigma-Aldrich, St. Louis, MO, USA, 97%) and 0.453 g (3.0 mmol) of 2-nitrobenzaldehyde (Sigma-Aldrich, 98%) in a small amount of absolute ethyl alcohol (Dinâmica LTDA) (3 mL) and using pulverized potassium hydroxide (JT Baker) as a catalyst. The mixture was stirred at room temperature (30 °C) until the precipitate formed (approximately five minutes). The reaction mixture was cooled and then filtered under reduced pressure. The obtained solid was recrystallized from absolute ethyl alcohol (Scheme 1).

**Scheme 1.** Synthesis of MNC.

### 2.2. Structure Characterization

X-ray diffraction data collection was performed using a Bruker APEX-II CCD diffractometer equipped with graphite-monochromated Mo Kα radiation at 120(2) K. Using Olex2 [24], MNC structure was solved using direct methods with SHELXT-2014 [25] program and refined using least-squares minimization with SHELXL-2014 [26] program. Hydrogens were constrained as riding atoms with $U_{iso}(H) = 1.2\ U_{eq}(C)$ for aromatic H and $U_{iso}(H) = 1.5\ U_{eq}(C)$ for H from methoxy group, with fixed C—H bond distances of 0.95 Å and 0.98 Å, respectively. Because it is involved in an intramolecular interaction, hydrogen H7 was freely refined. The chemical model was validated using Platon [27], whereas tables, figures, and images were generated by Olex2, Mercury [28,29], Ortep [30], and UCSF Chimera [31] programs. A conformational comparison was made through the Cambridge Structural Database (CSD) [32] with the Mogul program [33]. Crystal structure data are available free of charge in the Cambridge Structural Data Centre (CCDC) [34,35] under deposit number 2047630. Details of the refinement are provided in Table 1.

**Table 1.** Experimental details for single-crystal X-ray determination of MNC.

| | |
|---|---|
| Chemical formula | $C_{16}H_{13}NO_4$ |
| $M_r$ | 283.27 |
| Crystal system, space group | Monoclinic, $P2_1/c$ |
| Temperature (K) | 120 |
| $a, b, c$ (Å) | 12.4534 (19), 6.8905 (10), 15.555 (2) |
| β (°) | 101.834 (2) |
| $V$ (Å$^3$) | 1306.4 (3) |
| Z | 4 |
| Radiation type | Mo Kα |
| μ (mm$^{-1}$) | 0.11 |
| Crystal size (mm) | 0.32 × 0.23 × 0.22 |
| No. of measured, independent, and observed [$I > 2\sigma(I)$] reflections | 23943, 3233, 2703 |
| $R_{int}$ | 0.032 |
| $(\sin\theta/\lambda)_{max}$ (Å$^{-1}$) | 0.666 |
| $R[F^2 > 2\sigma(F^2)]$, $wR(F^2)$, $S$ | 0.037, 0.105, 1.04 |
| No. of reflections | 3233 |
| No. of parameters | 194 |
| H-atom treatment | H atoms treated with a mixture of independent and constrained refinement |
| $\Delta\rho_{max}$, $\Delta\rho_{min}$ (e Å$^{-3}$) | 0.34, −0.25 |

MNC single crystals were characterized using IR (Frontier—PerkinElmer, Waltham, MA, USA), $^1$H NMR (500MHz), $^{13}$C NMR (126MHz) (Avance III (Tesla)—Bruker, Billerica, Massachusetts, EUA), and GC-MS (QP2010 Ultra—Shimadzu, Kyoto, Japan). The GC-MS analyses were carried out using a CBP-5 capillary column (length 30 m, internal diameter 0.25 μm, and film thickness 0.25 mm). Manual injections with a volume of 1.0 μL were performed in Split mode (−1.0) as carrier gas helium gas (purity 99.999%) with a flow rate of 1.0 mL/min was used. The injector temperature was 280 °C, and the interface of the

ionization source was 250 °C. The initial oven temperature was 80 °C for the first 2 minutes, followed by a heating ramp of 40 °C/min to 280 °C, which was then maintained for 28 min.

**4′-methoxy-2-nitrochalcone:** $C_{16}H_{13}NO_4$ (283.27 g/mol), white crystal, yield 88%. **IR:** $UC=C$ (1523 $cm^{-1}$), $UC=O$ (1523 $cm^{-1}$), $UC-H$ $sp^2$ (3080 $cm^{-1}$), $UNO_2$ (1265 $cm^{-1}$). **$^1H$ NMR (500 MHz, CDCl$_3$):** δ 8.14–7.98 (m, 4H), 7.77–7.64 (m, 2H), 7.59–7.52 (m, 1H), 7.32 (d, J = 15.6 Hz, 1H), 7.05–6.95 (m, 2H), 3.89 (s, 3H). **$^{13}$C{H} NMR (126 MHz, CDCl$_3$):** δ 188.66 (s), 163.74 (s), 139.16 (s), 133.48 (s), 131.18 (s), 129.27 (s), 127.37 (s), 124.95 (s), 113.99 (s), 55.54 (s); **GCMS:** 237.09; 176.03; 148.04; 135.04; 107.05; 45.99.

### 2.3. Hirshfeld Surface Analysis

Hirshfeld surfaces are constructed by partitioning the electron density of a molecule or a crystal into individual atomic contributions. This partitioning is based on the concept of Hirshfeld weights, which assign portions of the total electron density to each atom in the system. These weights are determined by analyzing the distance between a reference point on the surface and the nuclei of the atoms in the molecule [36]. Among its properties, the *normalized contact distance* ($d_{norm}$) shows close contact regions on the surface, identified by red spots, and is given in Equation (1):

$$d_{norm} = \frac{d_i - r_i^{vdW}}{r_i^{vdW}} + \frac{d_e - r_e^{vdW}}{r_e^{vdW}} \tag{1}$$

where $d_i$ is the shortest distance between an inner atom and the surface and $d_e$ is the shortest distance between an outer atom and the surface. Both $d_e$ and $d_i$ are normalized based on their van der Waals radii of the correspondent atom, $r^{vdW}$ [37]. Also, the *shape index* (S) surface is a qualitative measurement useful in identifying nonclassical interactions by red and blue "bowtie" spots on it, following Equation (2):

$$S = \frac{2}{\pi}\arctan\left(\frac{\kappa_1 + \kappa_2}{\kappa_1 - \kappa_2}\right) \tag{2}$$

where $\kappa_1$ and $\kappa_2$ are the principal curvatures calculated in terms of principal directions $u$ and $v$ for each point on the surface [38]. Furthermore, the frequency of each combination of $d_i$ and $d_e$ was mapped using *fingerprint* plots, which show an overall sight of intermolecular interactions in crystal [39]. All Hirshfeld surfaces, graphics, and corresponding images were made with the CrystalExplorer17 program [40].

### 2.4. Theoretical Calculations

Considering that the cytotoxic assay is evaluated in aqueous media, all theoretical calculations were performed using $\varepsilon$ = 78.3553 (water solvent). Atom coordinates were taken from experimental single-crystal X-ray diffraction (SCXD) data. They were optimized using the exchange–correlation functional B3LYP [41] with a 6–311 g(d,p) basis set, a correct theory level for small organic molecules [42]. Theoretical calculations were carried out using Gaussian [43] software, and pictures were generated using both Gaussview [44] and Mercury [45] software. The nonconstraint optimization was confirmed to be in a local minimum analytic harmonic frequency. The molecular orbitals were also calculated at the same theory level, and the energy of the highest occupied molecular orbital (HOMO) and lowest unoccupied molecular orbital (LUMO) were estimated. Finally, the molecular electrostatic potential (MEP) map was also estimated.

### 2.5. Molecular Docking

The MCL-1 protein structure was obtained from the Protein Data Bank [46] database (ID: 6NE5 [47], 1.5 Å resolution). After downloading, the structure was imported into Maestro v.11.2 [48] software and prepared using the Protein Preparation Wizard tool. Hydrogen atoms were added with EPIK [49] (pH 7.8 ± 0.2) and minimized using the OPLS3 [50] force field. Then, the chemical structure of MNC was drawn using the program PICTO v. 4.4.0.4 [51]. The most stable tautomeric form and protonation state were then predicted

at pH 7.5 using the ligprep program from Maestro software v.11.2 [48]. Subsequently, a grid with dimensions of 21.98 Å $\times$ 2,47 Å $\times$ 31,04 Å (x, y, and z) and a volume of 9898 Å$^3$ was created around the binding site.

The selection of the docking pose and the number of docking poses considered were based on established protocols and standard practices in molecular docking [52]. In our study, we used the Glide program's high-resolution protocol within the Maestro software for molecular docking calculations. This protocol employs a sophisticated algorithm that explores multiple potential binding orientations and conformations of the ligand within the binding site of the protein.

During the docking process, Glide generates a set of docking poses ranked by their respective scores. The docking poses represent different potential binding modes of the ligand in the binding site. The scoring function, which considers factors such as protein–ligand interactions and energetics, is used to evaluate and rank the poses based on their predicted binding affinity. The docking pose with the lowest energy or the most favorable score is typically considered as the most plausible binding mode and serves as the representative pose in the analysis.

*2.6. Experimental Assays*

The cell line tested was the cancerous line HCT116 (histological type: colorectal carcinoma, origin: human, platelet concentration: $7 \times 10^4$ cells/mL) that shows MCL-1 protein expression, obtained through a donation by the National Cancer Institute of the United States (US-NCI). The cells were handled in a sterile vertical laminar flow chamber (VECO, model Biosafe 12, class II) and kept in a CO$_2$ incubator at 37 °C with a 5% CO$_2$ atmosphere (NUAIRE, model TS Autoflow). HCT116 tumor cells were cultured in 25 cm$^2$ cell culture bottles with a volume of 5 mL in RPMI 1640 medium (Gibco, New York, NY, USA), supplemented with 10% fetal bovine serum (FBS) and 1% antibiotic (penicillin/streptomycin).

Maintenance was carried out before the cells reached a confluence greater than 90% in the area. Cell growth was monitored daily using an inversion microscope (ZEISS, Axiovert 40C model). The medium was removed to maintain adhered cells, and the bottle was washed twice with sterile phosphate buffer solution (PBS). Trypsin-EDTA 0.5% (Gibco) was diluted ten times in the PBS buffer solution to suspend the cells. After being suspended, the action of trypsin was inhibited by the addition of medium supplemented with SBF. Part of the cells was removed from the bottle, and the volume was filled with complete medium.

2.6.1. Evaluation of Antiproliferative Activity In Vitro Using MTT Method

HCT116 cancer cells were plated in 96-well plates, with the plating concentration being $7 \times 10^4$ cells/mL. The 96-well plates were incubated in an oven at 37° C at 5% CO$_2$ for cell adhesion. Then, 100 µL of test sample diluted in the complete medium was added to the plates and incubated again for 72 h. The plates were centrifuged at 1500 rpm for 15 min, and the supernatant was removed. A total of 100 µL of MTT solution (0.5 mg/mL), diluted in RPMI 1640 or DMEM medium, was added to each well, and the plates were incubated for more than 3 h. After the incubation period, the plates were centrifuged at 3000 rpm for 10 min, and the supernatant was removed. The spectrophotometer (Beckman Coulter Inc., DTX-880 model) was used for reading at 595 nm. Formazan was resuspended in 100 µL of dimethylsulfoxide (DMSO) and added to each well. Multimode Detection Software (Beckman Coulter Inc., Brea, CA, USA) was used for this analysis. The absorbance values resulting from the tests with the MNC at 50 µM were transformed into a percentage of inhibition compared with the negative control. These data were analyzed based on the mean $\pm$ standard error of the triplicate experiments. As for the samples tested in serial dilution, the mean inhibitory concentration (IC$_{50}$) was determined with the respective confidence intervals (95% CI) obtained using nonlinear regression. The data analysis was performed using the program GraphPad Prism v.5.0.

### 2.6.2. Evaluation of Antiproliferative Activity Using the Trypan Blue Exclusion Method

For the test, the cell line HCT116 was plated in 24-well plates at $7 \times 10^4$ cells/mL and incubated for 24 h with MNC. Doxorubicin was used as a positive control at 0.5 μM. After the incubation periods, the cells were transferred to Eppendorf tubes and centrifuged for 5 min at 1500 rpm. The pellet was resuspended in 1 mL of PBS. In a 90 μL aliquot of cell suspension, 10 μL of 0.4% Trypan blue was added. Viable and nonviable cells were counted in a Neubauer chamber.

Data were analyzed based on the mean ± standard error of the triplicate. To verify the occurrence of a statistically significant difference between the groups, the data were compared using analysis of variance (ANOVA) followed by the Student–Newman–Keuls test for comparison between groups, with 5% significance ($p < 0.05$).

### 2.6.3. Morphological Analysis Using Differential Staining with Rapid Panoptic

HCT116 cells were plated in suspension in 24-well plates at $7 \times 10^4$ cells/mL and incubated 24 h with the selected compounds. Doxorubicin was used as a positive control at 0.5 μM, and the negative control was treated only with the vehicle (medium-plus cells HCT116). After the incubation periods, micrographs were made using an optical microscope, and then the cells were transferred to Eppendorf tubes and centrifuged for 5 min at 2000 rpm. The supernatant was discarded, and the cells were resuspended in 1 mL of PBS. The slides were prepared using a cytocentrifuge (model CT-2000, Center). The slides were then fixed with a 0.1% triarylmethane solution, and stained with a 0.1% xanthene solution and a 0.1% thiazine solution (Laborclin®, São Paulo, Brazil). The immersion time in each solution was approximately 5 sec. The slides were washed with distilled water to remove excess dye. After drying, they were mounted with coverslips fixed with Entellan® and visualized under an optical microscope.

The effect of MNC on HCT116 cells was observed through morphological analysis of the nucleus, cytoplasm, and plasma membrane using optical microscopy with the 20× objective (Olympus, Tokyo, Japan) with negative and positive controls.

## 3. Results and Discussion

### 3.1. Structural Analysis

MNC is a chalcone that presents an *E* configuration about the α,β-unsaturated carbonyl system, with a methoxy group substituted at the *para* position on one benzene ring and a nitro group at the *ortho* position on the other phenyl ring. It crystallized in the centrosymmetric monoclinic space group $P2_1/c$ with one molecule in the asymmetric unit and four per unit cell. The primary crystal data, data collection, and refinement information are presented in Table 1, followed by a thermal ellipsoid graphic with a 50% probability level of the asymmetric unit, presented in Figure 1.

The molecular structure of MNC is almost planar, presenting an angle between planes formed by aromatic rings A and B of 5.23°. Researchers relate the planarity of chalcones to their antifungal activity [53] and the intensity of second-order nonlinear optical emission [54]. Moreover, the methoxy group has an *anti*-conformation concerning the carbonyl group with $\omega_6 = 173.76°$. This exposes O4, allowing a C3—H3···O4 intermolecular interaction. Moreover, the nitro group exhibits $\omega_1 = 43.97°$ relative to the attached phenyl ring, a measurement comparable to 290 out of 7110 structures containing this fragment (4.08%) searched in CSD. This orientation allows for a conformation that minimizes side repulsion when interacting with other molecules, thus reducing the steric effect. Additional dihedral angles can be found in Table 2.

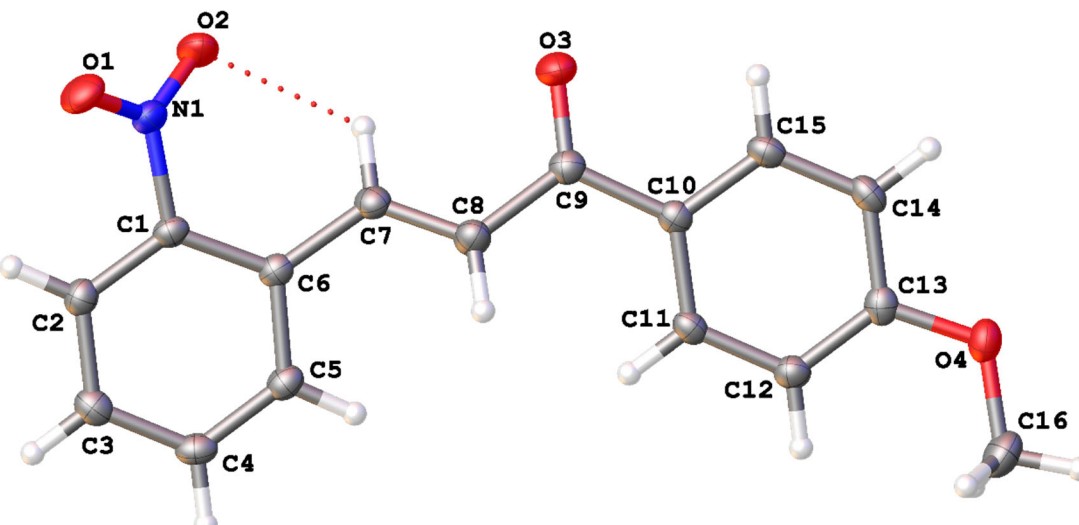

**Figure 1.** View of the title compound with the atom numbering scheme. Displacement ellipsoids for non-H atoms are drawn at the 50% probability level. Carbon, nitrogen, oxygen and hydrogen atoms are represented by gray, blue, red, and white, respectively. The red dashed line represents an intramolecular interaction.

**Table 2.** MNC selected dihedral angles.

| Dihedral Angle | Atoms | Value (°) |
|:---:|:---:|:---:|
| $\omega_1$ | O2—N1—C1—C6 | 44.0 (2) |
| $\omega_2$ | C1—C6—C7—C8 | 161.7 (1) |
| $\omega_3$ | C6—C7—C8—C9 | 174.8 (1) |
| $\omega_4$ | C7—C8—C9—C10 | 168.5 (1) |
| $\omega_5$ | C8—C9—C10—C15 | 167.9 (1) |
| $\omega_6$ | O3—C9—O4—C16 | 173.8 (1) |
| $\omega_7$ | C14—C13—O4—C16 | 172.7 (1) |

The crystal packing of MNC is stabilized mainly by C—H⋯O contacts with geometric parameters described in Table 3. C5—H5⋯O1 and C11—H11⋯O2 contacts assemble the molecule in a chain that runs along the [001] direction with a $C_2^2(12)$ motif. These chains are connected by C3—H3⋯O4 contacts to make sheets that coincide with the (010) plane, shown in Figure 2b. Figure 2c shows the C2—H2⋯O3 interactions that stack these layers along the [010] direction. The interatomic information for these contacts is given in Table 3.

**Table 3.** Primary intermolecular interaction geometry (Å, °) for MNC.

| D—H⋯A | D—H | H⋯A | D⋯A | D—H⋯A |
|:---:|:---:|:---:|:---:|:---:|
| C7—H7⋯O2 [i] | 0.97 (1) | 2.28 (1) | 2.8929 (15) | 121 (1) |
| C2—H2⋯O3 [ii] | 0.95 | 2.50 | 3.3750 (15) | 152 |
| C3—H3⋯O4 [iii] | 0.95 | 2.47 | 3.4179 (15) | 177 |
| C5—H5⋯O1 [iv] | 0.95 | 2.64 | 3.324 (15) | 129 |
| C11—H11⋯O2 [iv] | 0.95 | 2.42 | 3.2092 (15) | 141 |

Symmetry codes: ([i]) intramolecular; ([ii]) −x + 1, y − 1/2, −z + 1/2; ([iii]) x − 1, −y + 1/2, z − 1/2; ([iv]) x, −y + 1/2, z + 1/2.

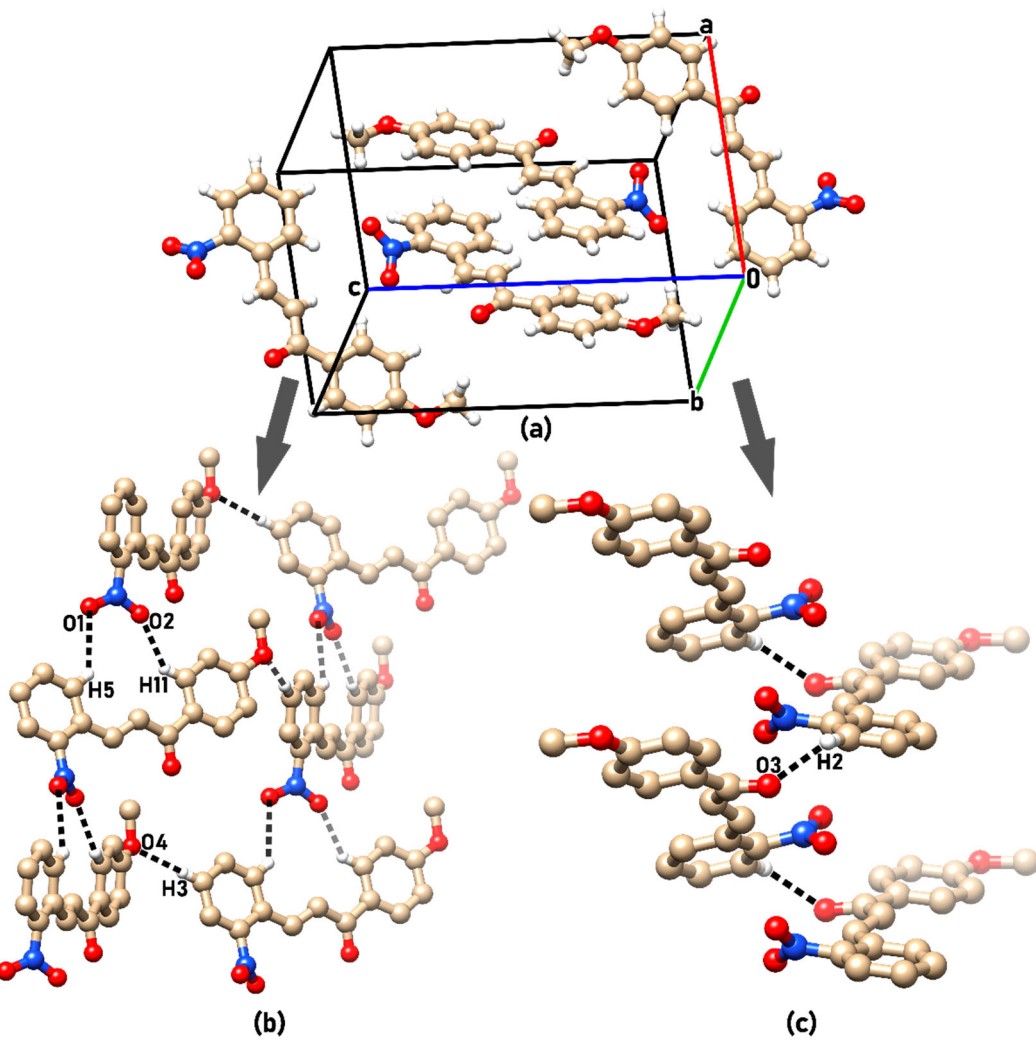

**Figure 2.** (**a**) Unit cell and packing of MNC. The crystal packing is stabilized by (**b**) C5—H5···O1, C11—H11···O2 and C3—H3···O4, and (**c**) C2—H2···O3 interactions.

The $d_{norm}$ Hirshfeld surface property shows a strong interatomic interaction for the C11—H11···O2 contact (red spot, Figure 3a), which is not observed for the C5—H5···O1 contact, as seen in Figure 3a. This suggests that among the interactions involving the nitro group, the former is dominant. Further, C—H···O interactions were confirmed through red spots in contact regions presented in Figure 3b and 3c. These plot regions are also mapped in the corresponding *fingerprint* plot in Figure 3d. Figure 4 shows high interaction points (red spots) suggesting nonclassical interactions between MNC molecules, which are recognized by the presence of regions with a red and blue "bow tie" in the *shape index* surface map. These interactions are quantitatively analyzed from their associated *fingerprint* plot. Figures 4c and 4f highlight the area regarding each element pair of contacts, finding C···C = 0.05%, C···H = 21.8%, C···O = 7.8%, H···H = 35.5%, and O···H = 28.8%, including reciprocal contacts.

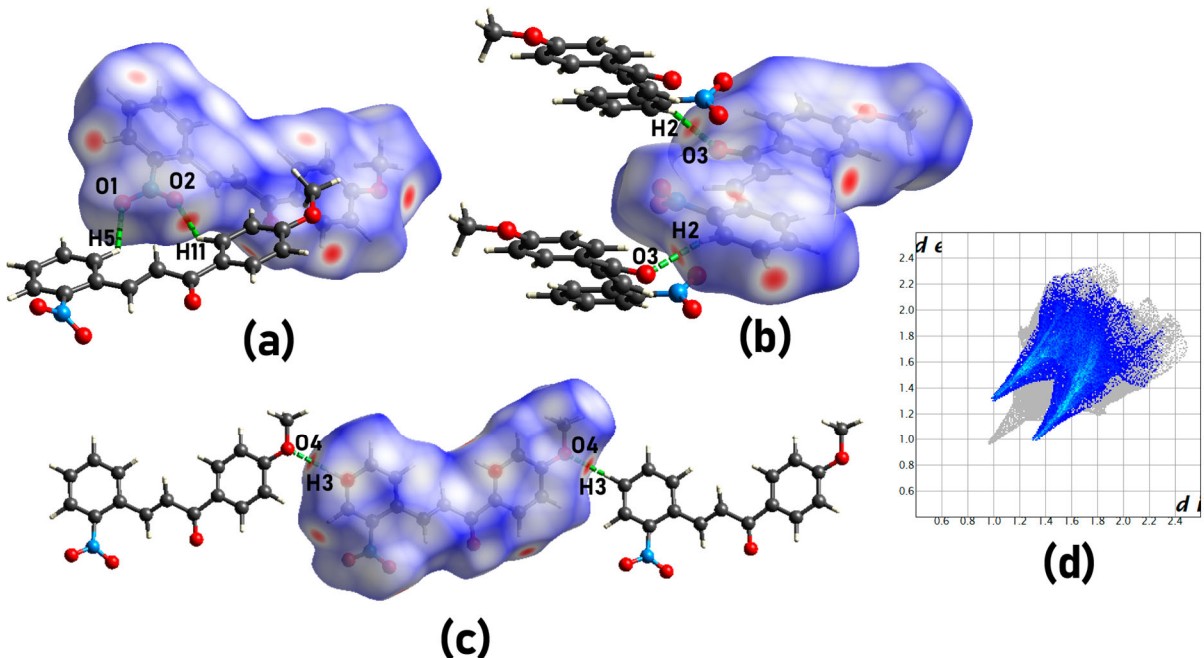

**Figure 3.** $d_{norm}$ graphic map of MNC highlighting (**a**) C11—H11···O2, (**b**) C2—H2···O3, and (**c**) C3—H3···O4 interactions. (**d**) O···H interactions mapped on the *fingerprint* plot.

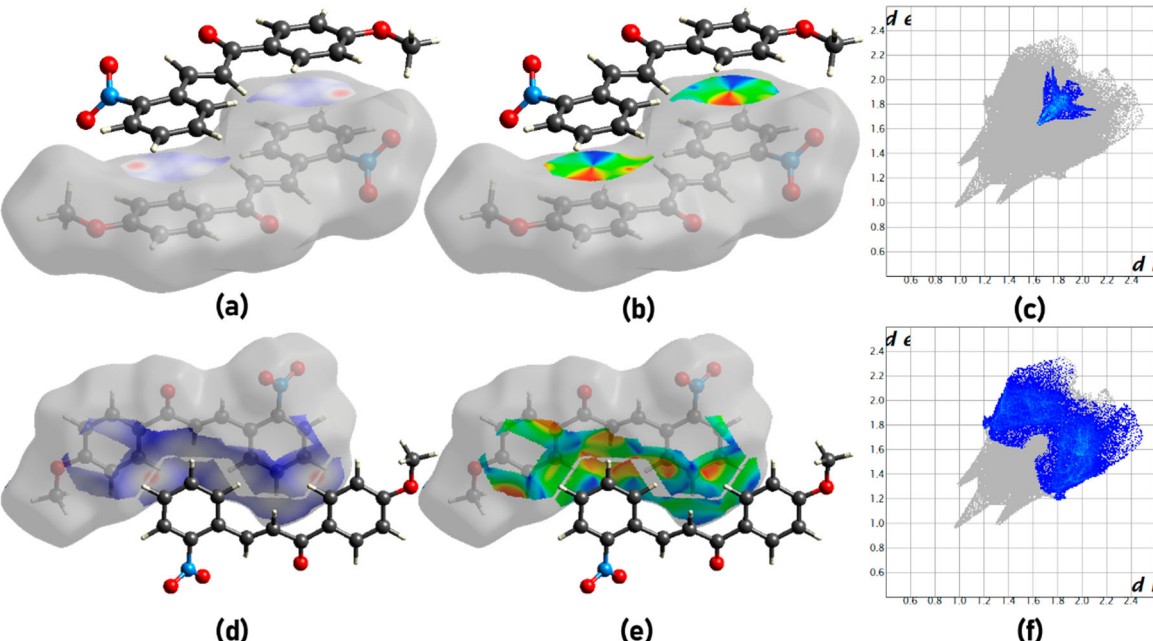

**Figure 4.** The surface area that corresponds to C···C contacts is highlighted in both $d_{norm}$ (**a**) and shape index (**b**) *surfaces*; (**c**) *fingerprint* plot, showing strong signs of $\pi \cdots \pi$ interaction between aromatic rings. Similarly, C···H contacts are represented on the $d_{norm}$ (**d**) surface as red spots between outside and inside surface aromatic rings. *The shape index surface (**e**) does not indicate the presence of C-H...$\pi$ interactions.* (**f**) $d_i$ and $d_e$ fingerprint highlighting C-H contacts.

### 3.2. Theoretical Calculations

The atomic coordinates of MNC obtained from SCXD were optimized using an aqueous medium simulation. Figure 5 shows an overlay between the molecular conformations experimentally found (blue) and theoretically calculated (cyan). Both the representation

in Figure 5 and the root-mean-square deviation (RMSD = 0.1620) show that, in a DMSO medium, MNC does not adopt a conformation so different from the solid state. The main difference found in these conformations is a slight rotation around C8–C9, which decreases the molecular planarity in DMSO.

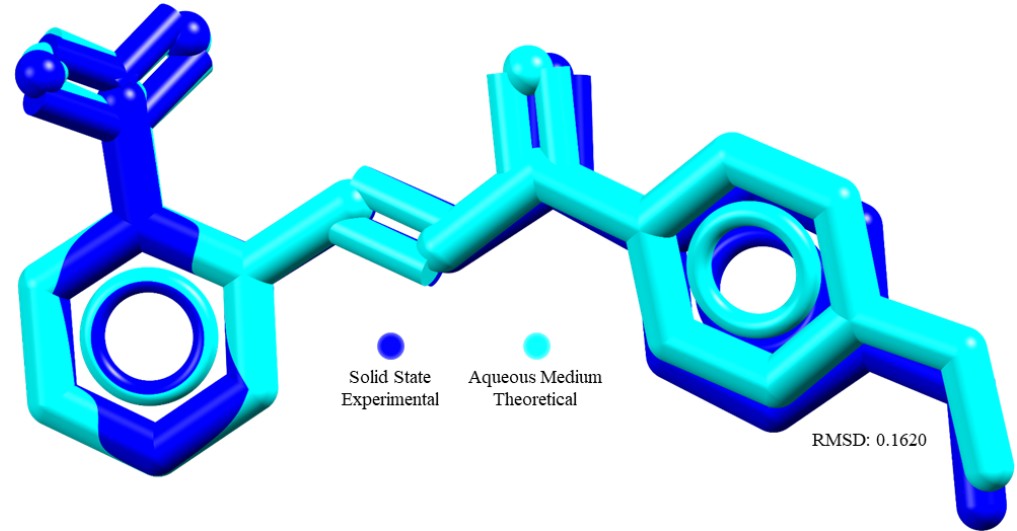

**Figure 5.** Overlay of the models of MNC in the solid state (SCXRD, blue) and aqueous medium (DFT-optimized, cyan).

The optimization found 548 molecular orbitals for MNC, and the representations of both HOMO and LUMO are presented in Figure 6a,b, respectively. The LUMO is spread over the nitrobenzene ring and carbonyl group, with energy $E_{LUMO} = -1.9862$ eV. On the other hand, the HOMO was found to be concentrated close to the methoxybenzene ring and olefin portion, with energy $E_{HOMO} = -7.9082$ eV. The HOMO–LUMO gap (difference in energies of these orbitals) is a descriptor of chemical reactivity in electron-transfer reactions. The greater the gap, the less reactive the molecule is [55]. Compared with other chalcones and their derivatives [56,57], MNC has high gap energy, with $E_{GAP} = 5.922$ eV. Finally, the MEP map calculated for MNC is presented in Figure 6c and shows the energy in different molecular sites as a color-based scale ranging from $\delta = -1.9862$ eV to $\delta = +1.9862$ eV. From this figure, it is possible to notice negative partial charges over the $O_{carbonyl}$ atom and positive partial charges over the $H_{olefin}$ atom.

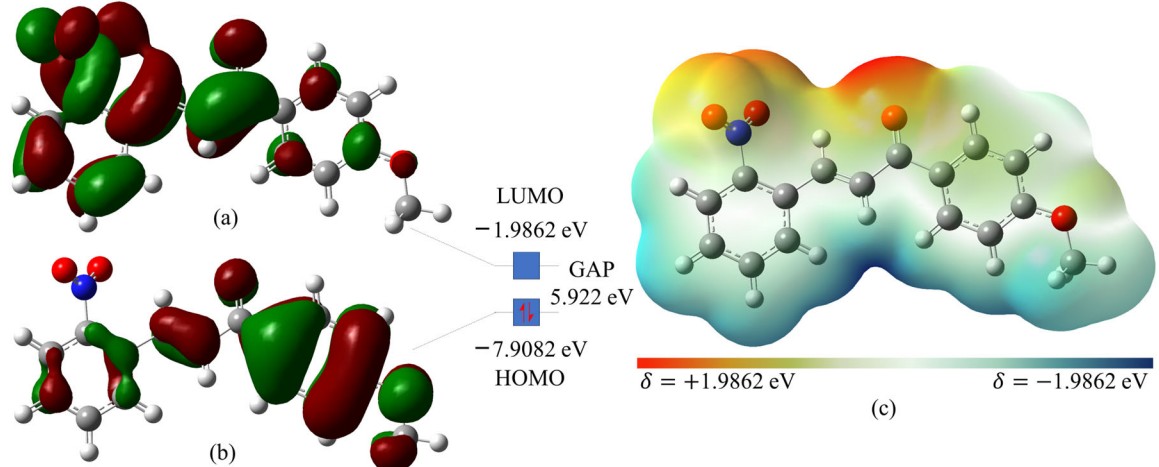

**Figure 6.** Representation of both (**a**) LUMO and (**b**) HOMO of MNC, with energy values for these orbitals and the gap between them; (**c**) MEP map for MNC.

### 3.3. Molecular Docking

Because chalcones have been found to inhibit the MCL-1 [20] protein, we investigated MNC as a new candidate for this purpose. The reference compound makes hydrogen bonds with Arg263 and Asn260, a π-π stacking interaction with Phe270, a water-mediated hydrogen bond interaction with Thr266, and hydrophobic interactions with Ala227, Phe228, Met231, Val253, Phe 254, Val258, Asn260, Arg263, and Leu267, with a Glide score of −15.212 kcal/mol. Table 4 shows the comparison of the Glide score between MNC and the reference ligand. MNC has suitable accommodation in the active site of the MCL-1 protein. It is supported by a π-π stacking observed between MNC and the phenylalanine residue (Phe270) and hydrophobic interactions with the valine and threonine residues (Val253 and Thr266, respectively) (Figure 7). The Glide score of MNC was −8.571 kcal/mol.

**Table 4.** Glide score comparison between the compound MNC and the cocrystallized ligand.

|  | Glide Score | RMSD |
| --- | --- | --- |
| 4′-methoxy-2-nitrochalcone | −8.571 | - |
| Redocking/reference ligand | −15.212 | 1.1 |

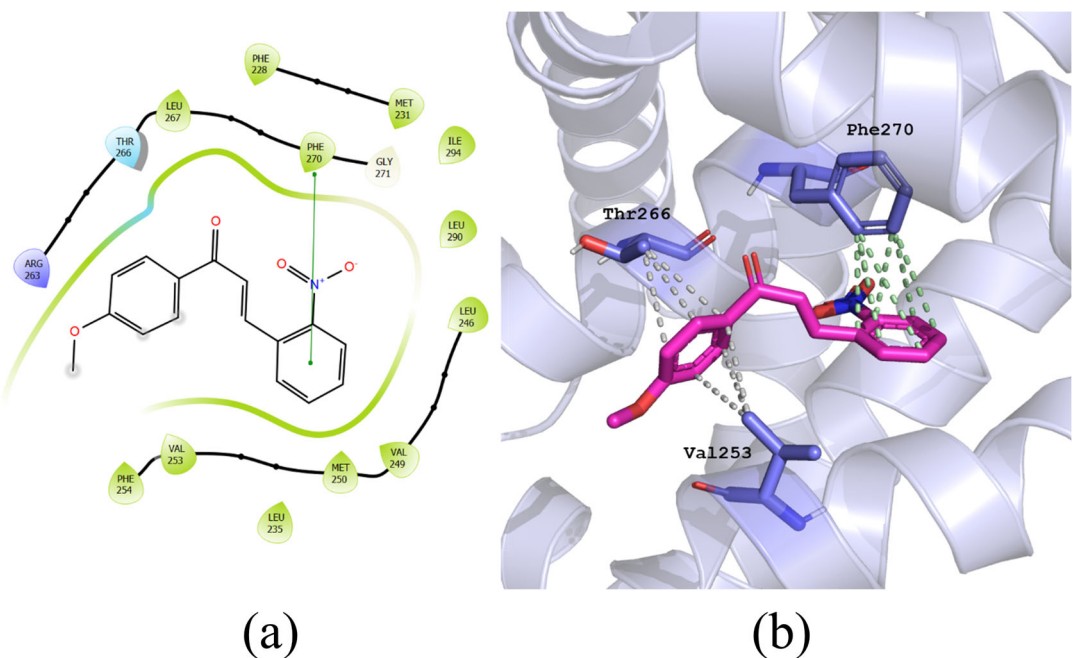

(a)                    (b)

**Figure 7.** Representation of (**a**) MNC molecular docking in 2D and (**b**) 3D.

The Blc-2 family proteins are located on the outer surface of the mitochondrial membrane. Their regular functions are regulated by protein–protein interactions, which makes their inhibition an inherent challenge. Mutagenesis approaches have been used to analyze the hot-spots of Bcl-2 family proteins and crystal ligand interaction analysis, revealing a conserved aspartic acid (Asp) and four hydrophobic residues (h1−h4) of proapoptotic proteins binding a conserved arginine (Arg) and four hydrophobic pockets (P1−P4) of antiapoptotic proteins, respectively [58]. That said, the compound we presented is essential, considering its binding affinity (Table 4). The π-π stacking interaction to the Phe270 residue constitutes the P2 hydrophobic pocket (Figure 7), relevant for MCL-1 inhibition [29]. Also, the hydrophobic interactions with Val253 and Thr266 are also crucial since these residues were considered hot-spots in the P2 and P3 pockets, respectively [58].

### 3.4. Cytotoxic Assay

MNC showed cell proliferation inhibition equal to 99% in the tumor line HCT116 in the test carried out with the concentration equivalent to 50 μM. The $IC_{50}$ values are shown in Table 5.

**Table 5.** $IC_{50}$ values for MNC concerning inhibition of HCT116 cell proliferation.

| $IC_{50}$ (μM)—24 h | $IC_{50}$ (μM)—48 h | $IC_{50}$ (μM)—72 h |
|---|---|---|
| 15.18 (14.21–16.23) | 8.69 (8.19–9.23) | 7.79 (7.54–8.05) |

The $IC_{50}$ obtained after 24 h was selected. We worked with three different concentrations: 15.18 μM ($IC_{50}$), 30.36 μM (2x $IC_{50}$), and 7.59 μM (1/2x $IC_{50}$) to evaluate cell viability using the Trypan blue exclusion method and for morphology analysis.

At a concentration of 30.36 μM, MNC reduces about 90% of viable cells' population and generates an unfeasibility of 25.72% of HCT116 cells. By decreasing the concentration, the number of viable cells was slightly recovered, but they are still significantly reduced regarding the negative control (Dox) (Figure 8).

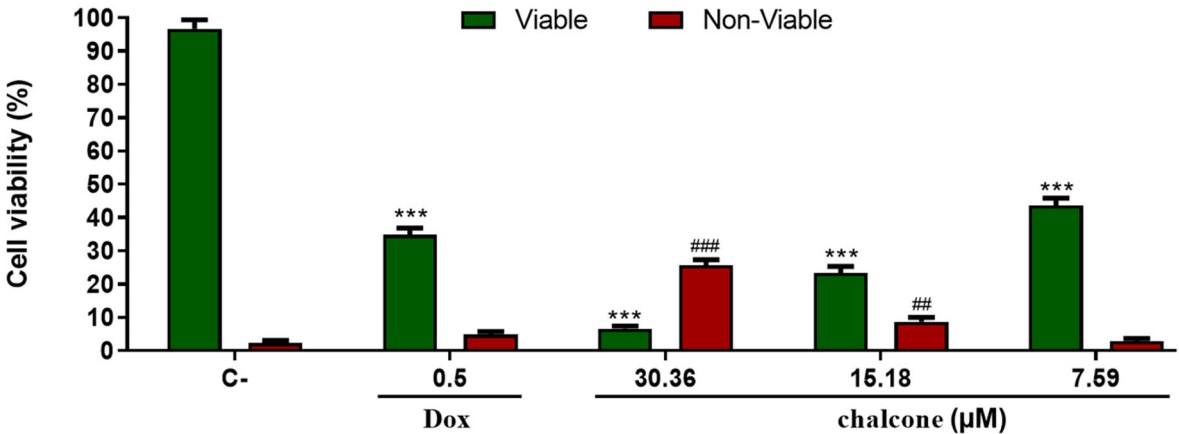

**Figure 8.** Effect of MNC on the viability of HCT116 determined using the Trypan blue exclusion assay after 24 h of treatment. Data are presented as mean percentages of viable and non-viable cells ± Standard Error of Mean (S.E.M.) of different concentrations of MNC (30.36, 15.18 and 7.59 μM) obtained from at least three independent experiments performed in triplicate. Doxorubicin (0.5 μM) was used as a positive assay control. One-way-ANOVA was performed, followed by Tukey's comparison test, where the P value was considered significant for *** ($p < 0.001$) when comparing the percentage of viable cells of the treated groups with the control negative and ### ($p < 0.001$)/## ($p < 0.003$) when comparing the percentage of non-viable cells in the treated groups with the negative control.

The treatment with 30.36 μM and 15.18 μM noticeably reduces cell density, as seen in the micrographs taken directly from the 24-well plate after the 24 h incubation with MNC (Figure 9). Compared with the negative control, the treated cells have a different morphology. At 7.59 μM, it is observed that the number of cells is reduced compared with the negative control and can be compared with the positive control (doxorubicin). The treatment stops the cycle cell in the G2 phase, causing the cell volume to increase.

Changes were observed both in cell number and in morphology in the micrographs obtained after incubation of MNC in HCT116 cells within 24 h. We infer that the action varies depending on the tested concentration. In contrast, very marked cytotoxicity is observed at concentrations of 15.58 μM and 7.79 μM, in which most of the cells have a ruptured plasma membrane and signs of apoptosis, respectively. At the concentration of 30.36 μM, MNC has a cytostatic antiproliferative action, where the cell number is reduced. This action is characterized by membrane "blebs" and also the slightly increased cell

volume ratio in the negative control, which can be compared with the group treated with doxorubicin (Figure 10).

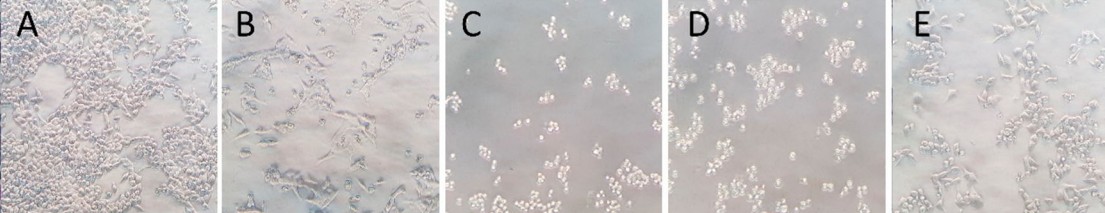

**Figure 9.** Images of the density of HCT116 cells after 24 h of treatment with MNC. Cells are visualized using optical microscopy. (**A,B**) Negative control and doxorubicin, respectively. (**C–E**) Cells treated at concentrations of 30.36; 15.18, and 7.59 μM, respectively.

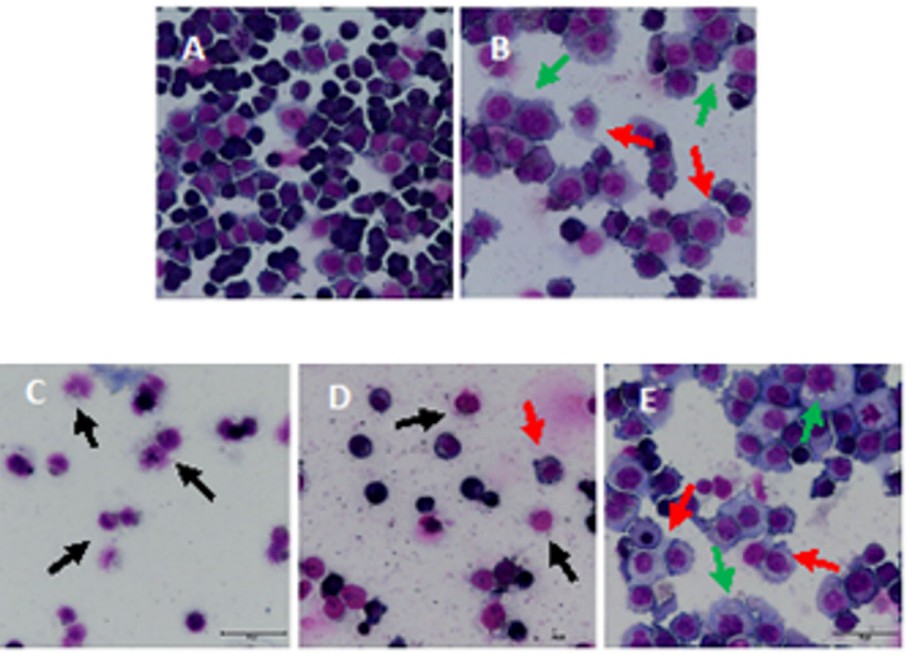

**Figure 10.** Morphology images of HCT116 cells after 24 h of treatment with MNC, stained with a rapid panoptic kit. Cells are visualized using optical microscopy. (**A,B**) Negative control and doxorubicin, respectively. (**C–E**) Cells treated at concentrations of 30.36, 15.18, and 7.59 μM, respectively. Bleeding cells are represented by red arrows; cells with increasing size are represented by green arrows; apoptosis/necrosis cells are represented with black arrows.

## 4. Discussion

The compound 4′methoxy-2-nitrochalcone (MNC) exhibits significant activity against HCT116 cells expressing MCL-1. It demonstrates a remarkable 99% inhibition of MCL-1 at a concentration of 50 μM, with an IC50 value of 15.18 μM within 24 h. Additionally, at concentrations of 15.58 μM, it displays notable cytotoxicity by inducing plasma membrane rupture in a majority of the cells, while at 7.79 μM, it exhibits signs of apoptosis. It is important to highlight that various small molecules have been developed to directly target MCL-1 and disrupt its antiapoptotic function. For example, S63845 and AZD5991 are potent MCL-1 inhibitors that bind to MCL-1 and trigger degradation of the protein. These inhibitors have shown promising results in preclinical studies and are being evaluated in clinical trials for the treatment of various cancers. Although these compounds show higher binding affinity towards MCL-1, MNC, as well as other natural products derived from plants, fungi, or marine organisms, can also be valuable in drug discovery and

development. Weak inhibitors can serve as starting points for further optimization and lead to the development of more potent compounds.

Molecular docking studies indicated that the compound exhibited a binding affinity within the active site of MCL-1. However, it is important to note that the docking score for the compound was found to be twice as high as that of the reference ligand, suggesting a potentially weaker binding affinity. Weak inhibitors and tight inhibitors play different roles in drug discovery and development, and their importance can vary depending on the specific context. In some cases, tight binding may not be desirable, especially when targeting dynamic or flexible binding sites. Weak inhibitors that can bind reversibly allow for more flexibility and accommodation of structural changes in the target protein, which can be beneficial in certain therapeutic contexts [59]. Additionally, tight inhibitors may have a higher risk of off-target effects and binding to unintended proteins due to their stronger interactions. Weak inhibitors, on the other hand, may exhibit better selectivity and have fewer off-target interactions, reducing the potential for adverse effects [60].

The binding affinity of MNC towards MCL-1 furthers its biological activity, particularly against MCL-1. The primary interactions identified include hydrogen bonding with Leu267 and $\pi \cdots \pi$ interactions with Phe270. These interactions are also observed in the crystal structure, primarily sustained by C—H$\cdots$O and $\pi \cdots \pi$ interactions. Notably, the compound possesses four potential hydrogen bond acceptor sites, which are confirmed by theoretical calculations. The theoretical calculations also suggest that the HOMO (highest occupied molecular orbital) is located around the nitro group, while higher values of MEP (molecular electrostatic potential) are mainly found around the nitro and carbonyl oxygens. The LUMO (lowest unoccupied molecular orbital) is situated around the aromatic ring with the methoxy group, indicating its electrophilic character. The compound's chemical stability is highlighted by its substantial energy gap of 5.922 eV.

Molecular docking is a static process and does not capture the dynamic nature of protein–ligand interactions. Ligand binding and interactions within the protein binding site can exhibit conformational changes and flexibility, which may affect the accuracy of the docking results. To address this concern, it is common practice to employ molecular dynamics simulations and other computational techniques to assess the stability and dynamics of the protein–ligand complex [61]. These techniques can provide valuable insights into the flexibility of the binding site, conformational changes, and the robustness of the predicted interactions.

In our study, we focused on the initial docking analysis to evaluate the binding affinity and potential interactions of the compound within the active site of the MCL-1 protein. We acknowledge that a single docking pose may not capture all possible binding modes and dynamic behaviors of the protein–ligand complex [61]. Therefore, further studies, including molecular dynamics simulations or experimental validation, would be essential to confirm and explore the stability and accuracy of the predicted binding interactions.

Regarding MCL-1 inhibition, the results suggest that an aromatic ring capable of engaging in $\pi \cdots \pi$ interactions, combined with hydrogen bond acceptor substituents, represents a reliable strategy in the search for other potential candidates. The chalcone core demonstrates its utility as a scaffold for the development of novel biologically active molecules. Chalcone-based compounds are relatively accessible and can be synthesized with diverse modifications, making them attractive candidates for drug development. Through optimization and further research, chalcone-based MCL-1 inhibitors have the potential to progress from preclinical studies to clinical trials, offering new therapeutic options for cancer patients.

The high concentrations of the compound used in the experiments (ranging from 7.59 μM to 30.36 μM) could suggest the need for a relatively high dose to achieve the desired bioavailability and target tissue concentration. However, it is important to note that this study primarily focused on evaluating the inhibitory effects of the compound on HCT116 cell proliferation and its interactions with MCL-1, rather than exploring its pharmacokinetic profile.

To address the concerns regarding ADME characteristics and dose-to-bioavailability ratio, further studies are necessary. These studies could investigate the compound's oral bioavailability, absorption, tissue distribution, metabolism, and elimination to determine its pharmacokinetic properties and potential suitability as a therapeutic agent. Additionally, optimization of the compound's structure or formulation strategies could be explored to enhance its bioavailability and reduce the required concentration for effective inhibition.

It is important to acknowledge that the presented study represents an initial step in exploring the potential of the novel small molecule as an antiproliferative agent. Subsequent research is needed to address the ADME considerations and fully assess its therapeutic potential in terms of pharmacokinetics, efficacy, and safety. Beyond ADME considerations, the development of chalcone-based MCL-1 inhibitors is an ongoing field of research, and further studies are needed to fully explore their therapeutic potential, safety profiles, and clinical efficacy. Nonetheless, the prospects for chalcone-based MCL-1 inhibitors offer exciting opportunities in the quest for improved cancer treatments.

**Author Contributions:** Conceptualization, J.M.F.C., H.B.N. and G.L.B.d.A.; methodology, L.F.N.N. and I.V.M.; software, B.J.N., J.V.B.B.; validation, J.M.F.C., H.B.N. and A.G.O.; formal analysis, J.M.F.C. and H.B.N.; resources, A.G.O. data curation, H.D.A.V. and J.E.Q.; writing—original draft preparation, L.F.N.N., I.V.M. and L.M.B. writing—review and editing, J.M.F.C., A.G.O., C.P., H.B.N. and G.L.B.d.A.; supervision, J.M.F.C., H.B.N. and G.L.B.d.A.; funding acquisition, H.B.N. All authors have read and agreed to the published version of the manuscript.

**Funding:** This research was funded by the Brazilian agencies Fundação de Amparo à Pesquisa do Estado de Goiás (FAPEG), Conselho Nacional de Desenvolvimento Científico e Tecnológico (CNPq), and Coordenação de Aperfeiçoamento de Pessoal de Nível Superior (CAPES, grant 001).

**Data Availability Statement:** The data presented in this study are available in the CCDC and supplementary information.

**Acknowledgments:** This research was developed with the support of the High-Performance Computation Nucleus of Universidade Estadual de Goiás. We are grateful to OpenEye Scientific Software Inc. (https://www.eyesopen.com/, accessed on 15 May 2023) for providing academic license of their program.

**Conflicts of Interest:** The authors declare no conflict of interest.

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
