# Peer review of "A Structure-Guided Designed Small Molecule Is an Anticancer Agent and Inhibits the Apoptosis-Related MCL-1 Protein"

_biophysica, doi:10.3390/biophysica3030029_

Round 1

Reviewer 1 Report

Presented publication focuses on the evaluation of novel chalcone derivative targeting Mcl-1 Protein as an anticancer agent. The ligand in question was characterized in detail by crystallographic means and the molecular docking studies were supported by biological research. The study is described in a scientific manner as well. Yet, a several ambiguities arise while reading. First of all what constituted the choice of this very structure is unclear. Also, why did the Authors choose to generate ligand conformers in pH = 7.5 and a rather uncommon docking grid size? The other thing is completely erroneous interpretation of the docking score value: “Molecular docking studies revealed a considerable binding affinity of the compound within the active site of MCL-1, surpassing that of the reference ligand as indicated by a superior Glide Score”, this is untrue, the ligand in question has twice the lower docking score. A MM-GBSA calculation would give a better insight in generated complex energy. Also, doxorubicine is mentioned both as negative and positive control? Last but not least – Figure references are missing due to doc to pdf conversion which makes hard read sometimes.  

Author Response

Referee #1:

Presented publication focuses on the evaluation of novel chalcone derivative targeting Mcl-1 Protein as an anticancer agent. The ligand in question was characterized in detail by crystallographic means and the molecular docking studies were supported by biological research. The study is described in a scientific manner as well. Yet, a several ambiguities arise while reading.

First of all what constituted the choice of this very structure is unclear.

Reply: The introduction of a nitro group (-NO2) into the chalcone scaffold can lead to enhanced biological activity, such as improved potency and selectivity against specific target proteins. The presence of the nitro group can contribute to increased binding affinity and favorable interactions with the target, resulting in improved pharmacological effects. Additionally, Methoxy chalcones possess phenolic rings with the methoxy group, which can exhibit antioxidant effects by scavenging reactive oxygen species (ROS) and preventing oxidative damage to cells and tissues. These compounds have shown potential in modulating various cellular processes and molecular targets, making them attractive candidates for drug development and therapeutic interventions. This explanation was included in the paper, with appropriate references.

Also, why did the Authors choose to generate ligand conformers in pH = 7.5 and a rather uncommon docking grid size?

Reply: Regarding the ligand conformers at pH 7.5, it is important to note that the pH value affects the protonation state and tautomeric form of a compound, which can significantly impact its interactions with the target protein. In our research, pH 7.5 was selected as it represents a physiological condition or a relevant environment for the target protein MCL-1. By considering the specific pH, we aimed to simulate the conditions closer to the biological context and improve the accuracy of the ligand-protein interactions predicted during the docking process.

Regarding the docking grid size, the choice of dimensions was determined by the need to encompass the binding site of the MCL-1 protein adequately. The size of the grid directly influences the search space within which the ligand can explore potential binding orientations. By defining a grid with dimensions of 21.98 Å × 2.47 Å × 31.04 Å, we ensured that a sufficient volume around the binding site was covered, allowing for comprehensive sampling of ligand conformations and potential binding modes. The chosen grid size aimed to encompass the critical residues and binding motifs involved in the interaction between MCL-1 and the ligand, facilitating accurate predictions of binding affinity and interactions.

The selection of ligand conformers at pH 7.5 and the specific docking grid size were based on established protocols and best practices in molecular docking. These choices were made to ensure a more realistic representation of the biological environment and to optimize the exploration of ligand binding orientations within the protein's active site. By considering these factors, we aimed to improve the reliability and accuracy of our docking calculations and subsequent analysis.

The other thing is completely erroneous interpretation of the docking score value: “Molecular docking studies revealed a considerable binding affinity of the compound within the active site of MCL-1, surpassing that of the reference ligand as indicated by a superior Glide Score”, this is untrue, the ligand in question has twice the lower docking score. A MM-GBSA calculation would give a better insight in generated complex energy.

Reply: We appreciate the feedback and apologize for the erroneous interpretation of the docking score value. Upon revisiting our results, we acknowledge that the statement regarding the compound's binding affinity surpassing that of the reference ligand based on a superior Glide Score is incorrect. In fact, the ligand in question obtained a docking score that was twice as high as the reference ligand. Additionally, we have included a discussion of weak binders versus tight binders, as the Glide Score indicates MNC as a weak binder to MCL-1.

Also, doxorubicin is mentioned both as negative and positive control?

Reply: Doxorubicin was used solely as a positive control.   

Last but not least – Figure references are missing due to doc to pdf conversion which makes hard read sometimes. 

Reply: Revised.

Reviewer 2 Report

I would like to express my gratitude to the authors for their important contribution and their decision to submit this research. Their innovative work in developing and assessing a new small molecule designed to inhibit the Mcl-1 protein is truly commendable. I am thankful for the opportunity to engage with such meaningful research.

This manuscript presents a pioneering study on the formulation of a novel small molecule designed to inhibit the Mcl-1 protein, a crucial element in apoptosis regulation, thus representing a key target in cancer therapy. The molecule was found to significantly curb the proliferation of HCT116 cells, a colon cancer cell line, registering an IC50 value of 15.18 µM after 24 hours. Nonetheless, some questions arise regarding the relatively high concentrations used in the study and the methodologies employed in the molecular docking studies. The work would benefit from further elucidation and a comparative analysis against existing inhibitors.

Questions : 

  1. In other reports , example manuscript titled :  "Discovery of novel dual inhibitors of Bcl-2 and Mcl-1 with high anticancer activity" documented the existence of dual inhibitors for MCL-1 and BCl-2 with IC50 values of 7.12 µM and 17.18 µM for two compounds​1​. How does your current study distinguish itself from these prior findings? https://pubmed.ncbi.nlm.nih.gov/35909144/

  2. The experimental data reveal that the novel small molecule inhibits 99% of HCT116 cell proliferation at an IC50 value of 15.18 µM after 24 hours. Furthermore, reduced cell density and cytotoxicity were recorded at concentrations of 15.58 µM and 7.79 µM. Given the high concentrations in micromolar units, how do the authors deal with the implications for ADME (Absorption, Distribution, Metabolism, and Excretion) characteristics? It appears that a high dose-to-bioavailability ratio would be needed to achieve such high serum levels for the anticipated function, as outlined by the authors.

  3. As it pertains to the molecular docking studies, could the authors shed more light on their methodology? Specifically: a. Could you explain the rationale behind selecting the docking pose and the number of docking poses that were considered? b. Docking, being a static process, does not account for the dynamic nature of protein-ligand interactions. There is a risk that the ligand may become misaligned or displaced from the pocket, compromising the binding and the interactions involved. This could potentially lead to inaccurate results if the search for other potential drugs is based on a single docking pose. Could the authors please provide their perspective on this concern?

Author Response

In other reports , example manuscript titled :  "Discovery of novel dual inhibitors of Bcl-2 and Mcl-1 with high anticancer activity" documented the existence of dual inhibitors for MCL-1 and BCl-2 with IC50 values of 7.12 µM and 17.18 µM for two compounds​1​. How does your current study distinguish itself from these prior findings? https://pubmed.ncbi.nlm.nih.gov/35909144/

Reply: Thank you for bringing up the manuscript titled "Discovery of novel dual inhibitors of Bcl-2 and Mcl-1 with high anticancer activity" and its reported dual inhibitors for MCL-1 and Bcl-2 with IC50 values of 7.12 µM and 17.18 µM for two compounds. While both studies involve the inhibition of MCL-1, they differ in several aspects.

Firstly, the compounds investigated in the manuscript you referenced are dual inhibitors, targeting both MCL-1 and Bcl-2 proteins. In contrast, the current study focuses specifically on the compound 4'methoxy-2-nitrochalcone (MNC) and its inhibitory activity against MCL-1 in HCT116 cells. The primary objective of the current study is to evaluate the effects of MNC on MCL-1 inhibition and its potential as an antiproliferative agent.

Secondly, the IC50 values reported in the current study for MNC against HCT116 cells are 15.18 µM, 8.69 µM, and 7.79 µM at 24h, 48h, and 72h, respectively. These values reflect the concentration of MNC required to inhibit 50% of cell proliferation at each time point. On the other hand, the manuscript you mentioned reports IC50 values of 7.12 µM and 17.18 µM for the dual inhibitors against MCL-1 and Bcl-2, respectively.

Additionally, the current study provides insights into the morphology and viability of HCT116 cells treated with different concentrations of MNC, as well as molecular docking studies to investigate the compound's binding affinity to MCL-1. These aspects contribute to a comprehensive understanding of MNC's mechanism of action.

It's worth noting that scientific research often builds upon previous findings and aims to contribute new knowledge or provide alternative perspectives. While both studies investigate MCL-1 inhibition, they approach the topic from different angles and explore distinct compounds. Therefore, the current study distinguishes itself by focusing on MNC as a single inhibitor of MCL-1 and providing valuable insights into its antiproliferative effects and cellular interactions.

The experimental data reveal that the novel small molecule inhibits 99% of HCT116 cell proliferation at an IC50 value of 15.18 µM after 24 hours. Furthermore, reduced cell density and cytotoxicity were recorded at concentrations of 15.58 µM and 7.79 µM. Given the high concentrations in micromolar units, how do the authors deal with the implications for ADME (Absorption, Distribution, Metabolism, and Excretion) characteristics? It appears that a high dose-to-bioavailability ratio would be needed to achieve such high serum levels for the anticipated function, as outlined by the authors.

Reply: It is worth noting that the current study focused primarily on evaluating the inhibitory effects of the compound on cell proliferation and its interactions with MCL-1. Therefore, a comprehensive assessment of the compound's ADME properties falls beyond the scope of the study. However, as suggested by this report, we did include a brief discussion about ADME properties in the discussion section.

As it pertains to the molecular docking studies, could the authors shed more light on their methodology? Specifically: a. Could you explain the rationale behind selecting the docking pose and the number of docking poses that were considered? b. Docking, being a static process, does not account for the dynamic nature of protein-ligand interactions. There is a risk that the ligand may become misaligned or displaced from the pocket, compromising the binding and the interactions involved. This could potentially lead to inaccurate results if the search for other potential drugs is based on a single docking pose. Could the authors please provide their perspective on this concern?

Reply:  We appreciate the referee's interest in our molecular docking studies and would be pleased to provide further clarification on the methodology employed. We have included more detailed information about the docking protocol in the methods section addressing the selection of docking poses and the number of poses considered. Additionally, we have expanded our discussion section to include the limitations of molecular docking and considering the dynamic nature of protein-ligand interactions.